# The Diversification and Enhancement of an IDS Scheme for the Cybersecurity Needs of Modern Supply Chains

Dimitris Deyannis [1,2,3], Eva Papadogiannaki [2,3,4], Grigorios Chrysos [4,*], Konstantinos Georgopoulos [4,*] and Sotiris Ioannidis [5]

1. Sphynx Technology Solutions AG, 6300 Zug, Switzerland; d.ntegiannis@sphynx.ch or deyannis@csd.uoc.gr
2. Foundation for Research and Technology-Hellas, Institute of Computer Science, GR-70013 Heraklion, Greece; epapado@ics.forth.gr or papadogian@csd.uoc.gr
3. Computer Science Department, University of Crete, GR-70013 Heraklion, Greece
4. Telecommunication Systems Research Institute, GR-73100 Chania, Greece; epapadogiannaki@tsi.gr
5. School of Electrical and Computer Engineering (ECE), Technical University of Crete, GR-73100 Chania, Greece; sotiris@ece.tuc.gr
* Correspondence: chrysos@mhl.tuc.gr (G.C.); kgeorgopoulos@tsi.gr (K.G.)

**Abstract:** Despite the tremendous socioeconomic importance of supply chains (SCs), security officers and operators are faced with no easy and integrated way for protecting their critical, and interconnected, infrastructures from cyber-attacks. As a result, solutions and methodologies that support the detection of malicious activity on SCs are constantly researched into and proposed. Hence, this work presents the implementation of a low-cost reconfigurable intrusion detection system (IDS), on the edge, that can be easily integrated into SC networks, thereby elevating the featured levels of security. Specifically, the proposed system offers real-time cybersecurity intrusion detection over high-speed networks and services by offloading elements of the security check workloads on dedicated reconfigurable hardware. Our solution uses a novel framework that implements the Aho–Corasick algorithm on the reconfigurable fabric of a multi-processor system-on-chip (MPSoC), which supports parallel matching for multiple network packet patterns. The initial performance evaluation of this proof-of-concept shows that it holds the potential to outperform existing software-based solutions while unburdening SC nodes from demanding cybersecurity check workloads. The proposed system performance and its efficiency were evaluated using a real-life environment in the context of European Union's Horizon 2020 research and innovation program, i.e., CYRENE.

**Keywords:** cybersecurity; supply chain services; network intrusion detection; pattern matching; reconfigurable parallel computing; network packet processing

## 1. Introduction

A supply chain, Figure 1, is a connected system of organizations, activities, information, and resources designed to source, produce, and move goods from suppliers to end customers. Modern SCs are a network of IT infrastructures and technologies that are used to connect, build, and share data [2], inadvertently giving rise to new forms of risk since they are not connected to physical products or physical locations. Despite the high socioeconomic importance of SCs, there is no straightforward way to protect their infrastructures from cyber threats, which are becoming increasingly prevalent and sophisticated in their execution [3]. Thus, SCs are in a state of constant refinement of their security capabilities in order to address their vulnerabilities [4–6].

**Figure 1.** Supply chain components.

### 1.1. Supply Chain and Cybersecurity

Currently, there exists no established cybersecurity method that guarantees the complete prevention of SC attacks [7]. Common risks are data breaches and corporate fraud originating from ransomware attacks and malware infections [8]. A solution to these issues is the use of tools that promptly identify and provide alerts about malicious activity in a supply chain infrastructure and, subsequently, limit the impact of the attack [9]. Such type of tools are the intrusion detection systems (IDSs) [10–12] and the intrusion prevention systems (IPSs) [13,14].

Specifically, intrusion detection systems are software applications or hardware devices that detect vulnerability exploits, malicious activity, or policy violations. IDS tools detect threats on network devices, such as firewalls on servers and routers, and send alerts to the corresponding system security department when an intrusion attempt takes place. On the other hand, IPS tools define specific actions when malicious activity is detected, i.e., they block incoming traffic from specific infected nodes or network packets that contain a specific type of information.

Moreover, intrusion detection systems are split into two different groups based on the events that they examine and the place that they are installed. The host-based intrusion detection systems (HIDSs) examine security events that take place on specific network nodes, while the network-based intrusion detection systems (NIDSs) are placed as intermediate network nodes and examine the traffic on the network. Additionally, the typical NIDSs include packet sniffers that gather network traffic for analysis based on specific rules. NIDSs can be installed on dedicated hardware platforms, which can examine network traffic independently.

Snort is one of the most popular NIDS methods used in the industry, since it is an open-source tool supported and updated by CISCO [10,15]. Snort runs in three different modes so that it can be used either as an intrusion prevention and/or as an intrusion detection framework. The available modes are (i) sniffer mode, where the system just sniffs the packets and displays packet-relevant information, (ii) packet logger mode, where the algorithm logs packets in log files, and (iii) intrusion detection mode, where the framework uses a set of rules to inspect packets and takes action in relation to them. Snort uses a multiple pattern search framework, i.e., the Aho–Corasick algorithm, and matches the incoming network packets with all signatures, concurrently. In fact, the capabilities featured by the Snort IDS have turned it into one of the strongest and most popular IDS tools embraced by a huge user community [16]. Nonetheless, recent cybersecurity reviews have shown that it is faced with slow packet processing rates [17], which stand as the main obstacle for its adoption as a high-end generic IDS solution.

### 1.2. Challenges

Due to the diversity of modern cybersecurity and privacy threats, security systems are pitched against the following three unprecedented challenges.

- Efficient cybersecurity services over complex supply chain networks and processes

The significant complexity of SC infrastructures, coupled with the corresponding SC operations, technologies, and mechanisms, poses critical performance requirements, where all the processes strive toward complete automation. On the other hand, the high complexity of these processes and the low network security increase the risk for cyber-attacks. Based on this, and according to ENISA's (European Union Agency for Cybersecurity) [18] direc-

tions, modern supply chains need to include cybersecurity services which meet demanding performance expectations while offering constant and vigorous monitoring.

- Offloading of demanding cybersecurity workloads from the main supply chain processing units onto dedicated processing platforms

Most of the monitoring and intrusion detection systems are software-based and reside on the SC network nodes. Despite the fact that these software-based solutions are flexible and can be easily programmed, they require a relative long time to process incoming streams of network packets. On the other hand, the assets, where the cybersecurity solutions are mapped, need to run their main workload concurrently, leading to either packet loss or low processing performance. Based on this, new independent components need to be integrated into the supply chain infrastructure in order to unburden the assets from having to support cybersecurity checks. In addition, these new solutions need to offer high processing and networking performance in order to be easily integrated and without any performance delays on the SC.

- Continuous and dynamic updating of the cybersecurity systems

The existing SC security systems offer threat prevention and mitigation techniques based on static network intrusion detection rules. On the other hand, the fast increase of new SC attacks [19], e.g., supply chain attacks are expected to increase by a factor of four each year [20], and the continuous updates of IDS databases [21] due to emerging cyber-attacks, lead to the need for novel cybersecurity systems that can be updated continuously and in a dynamic way. That is, new approaches need to facilitate the coordinated, continuous, and dynamic updating of cybersecurity information in real time. Additionally, the updates need to be handled automatically by the SC framework; thus, they need to be applied concurrently to all SC assets. This direction increases the SC update complexity due to the huge variety of assets.

### 1.3. Contributions

Taking into consideration the challenges listed above, this paper introduces an IDS solution that resides on a high-end, low-cost, and low-resources reconfigurable node suitable for supply chain infrastructures. In addition, it is a solution that can be completely offloaded on dedicated hardware (the FPGA of an MPSoC), thereby completely disentangling the supply chain assets from cybersecurity workloads while supporting dynamic updating of the security features. Specifically, the main contributions of this work are the following:

- The introduction of a novel SC reconfigurable-based cybersecurity technology. The proposed solution strengthens the SC network security and unburdens the network from the responsibility of IDS checks, which can now take place inside custom reconfigurable hardware boards.
- The complete integration and performance optimization of the most popular IDS software, i.e., Snort, on a low-cost and low-resources reconfigurable platform, i.e., the PYNQ Z1 development board. Additionally, this work is the first that integrates such small reconfigurable modules into complex and modern supply chain infrastructures.
- The combination of a multi-pattern matching algorithm, i.e., Aho–Corasick, with novel data structures that offer high computational power. The novel data structures are efficiently mapped on reconfigurable technology, taking advantage of hardware parallelism.
- The technology and its implementation platform are fully integrated with a real-life SC infrastructure, i.e., the CYRENE [1] paradigm, while real-life data processing workloads are used for their evaluation. The performance achieved shows that the proposed solution can be easily integrated with any complex supply chain environment and can host the complete offloading of the IDS workload using reconfigurable hardware.
- An energy consumption comparison between an IDS-software-based approach and our MPSoC/FPGA-based implementation using real-world data mining workloads.

### 1.4. Outline

The rest of this manuscript is organized as follows: Section 2 presents previous GPU- and FPGA-based work on the parallel pattern matching workload. Additionally, this section describes previous work that combined reconfigurable technology with supply chain infrastructures. Section 3 describes the novel Aho–Corasick algorithm implementation and its integration into the Snort framework. Section 4 offers a description of the implemented system. Section 5 evaluates the performance of our low-cost system when handling real-world SC infrastructure data. Finally, in Section 6, we comment on the proposed technology and discuss future directions, while Section 7 presents the paper's conclusions.

## 2. Related Work

This section presents previous work that focuses on mapping cybersecurity algorithms onto hardware-based platforms, i.e., GPUs and FPGAs, as well as work where the reconfigurable technology is integrated with the supply chain framework.

### 2.1. GPU-Based Pattern Matching

With the advent of commodity hardware resources in the market, many published works use high-end accelerators, such as GPUs, to boost the processing performance of typical network packet processing applications. For instance, Gnort [22] accelerates the pattern matching engine of the Snort IDS using a discrete GPU. Similarly, Kargus [23] performs load balancing in pattern matching workloads and is compatible with Snort IDS. MIDeA [24] offers a multi-parallel intrusion detection architecture tailored to multi-queue NICs, multiple CPUs, and multiple GPUs. Snap [25] and GASPP [26] are frameworks for programmable network traffic processing that can simplify the development of GPU-accelerated workloads. DFC [27] offers accelerated string matching tailored to packet processing by reducing memory accesses and cache misses. Moreover, in HeaderHunter [28,29], the authors propose the utilization of GPUs to accelerate network packet metadata processing for intrusion detection in encrypted communications, by converting the Aho–Corasick algorithm to match sequences of unsigned short numbers instead of characters. Furthermore, Papadogiannaki et al. [30] proposed a scheduling approach that, based on performance policies (such as high throughput or low power consumption), determines the most suitable combination of heterogeneous devices (i.e., CPU, integrated, or discrete GPUs) for efficient execution of network packet processing workloads (such as DPI or packet encryption). Similarly, in Pythia [31,32], authors add the support for concurrent execution of different network packet processing applications across multiple and heterogeneous devices. In APUNet [33], authors propose the utilization of integrated GPUs to accelerate packet processing workloads without paying the overheads of memory transactions between the host and discrete GPUs. Finally, in NBA [34], authors extend the Click router to leverage hardware accelerators for load balancing typical packet processing.

Unlike these works, we choose to accelerate the core operation of a network intrusion detection system, which is part of a supply chain, using a small FPGA platform due to its processing and power consumption characteristics (i.e., low cost and low resources).

### 2.2. FPGA-Based Pattern Matching

Many pattern matching and intrusion detection applications have been efficiently mapped on reconfigurable hardware. Sourdis et al. [35], who were the first, mapped a network intrusion detection application on a reconfigurable platform with impressive performance results. Song et al. [36] presented a packet classification architecture, i.e., BV-TCAM, on an FPGA-based network intrusion detection system (NIDS) offering multiple packet matches at gigabit per second network link rates. Baker et al. [37] were the first to propose an automatic tool that creates efficient mapping of intrusion detection algorithms using system-level optimizations. Das et al. [38] proposed an FPGA-based architecture for anomaly detection in network transmissions using a feature extraction module (FEM). Pontarelli et al. [39] proposed the implementation of a traffic-aware approach in the design

of FPGA-based NIDSs. Specifically, their framework splits the traffic into homogeneous groups of packets to different hardware blocks, which support smaller rule sets. Kim and Park [40] proposed an FPGA-based NIDS for detecting malicious network packets within messages of the IEC 61850 type. Zhao et al. [41] presented the novel Pigasus IDS/IPS tool, which is the first work where the majority of processing and all state and control flow are managed on an FPGA device. Their experiments show that Pigasus can support up to 100 Gbps using an average of five cores and one FPGA and consuming $38\times$ less power than a CPU-only approach. Le Jeune et al. [42] implemented the first FPGA-based convolution neural network for NIDSs. Their results show that their implementation needs further optimization in order to manage real-time intrusion detection. Finally, Ngo et al. [43] proposed a versatile framework for real-time Internet of Things (IoT) network intrusion detection using an artificial neural network (ANN) on a heterogeneous hardware platform.

In comparison to all the work performed so far, the technology introduced here is the first to develop an efficient NIDS solution that is hosted on a low-cost and low-resources reconfigurable platform (PYNQ Z1 development board). Specifically, the proposed system is the first FPGA-based system that can be seamlessly integrated with a real-life supply chain infrastructure offering high-performance cybersecurity services while completely offloading the security-related functionality from the main SC processing elements.

### 2.3. Supply Chain Framework and FPGAs

Overall, supply chains are a relatively young research area and, as a result, not much related work exists on the combination of reconfigurable technology with supply chain architectures. Zhou et al. [44] used FPGA devices for logistics supply chain information processing in order to reduce the costs and improve the services for end-users. Zou et al. [45] proposed an FPGA-based framework that accelerated the supply chain data processing workload and reduced the risk for time-delay processing. Finally, Ting Li [46] proposed a supply chain management framework that combined a neural network framework based on reconfigurable devices. Crucially, the SC/FPGA research published so far does not address the combination of security algorithms for SCs with reconfigurable hardware. This is the gap that the work presented here is aiming to fill.

Table 1 presents the most recent and relevant works with respect to (i) the hardware accelerator used, (ii) the testing condition, (iii) the application, and (iv) the evaluation/performance goals. The testing condition indicates whether the tool presented in each work was evaluated as part of a supply chain infrastructure or not.

**Table 1.** Comparison of recent and closely related works.

| Work | Hardware | | In SC | App | Goal | | |
|---|---|---|---|---|---|---|---|
| | GPU | FPGA | | | Exec. Time | Cost | Power |
| GASPP [26] | ✓ | – | – | Network packet | ✓ | ✓ | – |
| Pythia [32] | ✓ | – | – | Network packet processing | ✓ | – | ✓ |
| Kim et al. [40] | – | ✓ | – | IDS | ✓ | – | ✓ |
| Zhao et al. [41] | – | ✓ | – | IDS/IPS | ✓ | ✓ | ✓ |
| Le Jeune et al. [42] | – | ✓ | – | IDS | ✓ | – | – |
| Ngo et al. [43] | – | ✓ | – | IDS | ✓ | ✓ | – |
| Zhou et al. [44] | – | ✓ | ✓ | Logistics | ✓ | ✓ | – |
| Zou et al. [45] | – | ✓ | ✓ | SC traffic | ✓ | – | – |
| Ting Li [46] | – | ✓ | ✓ | Network manag. | ✓ | ✓ | ✓ |
| *This work* | – | ✓ | ✓ | IDS | ✓ | ✓ | ✓ |

In short, both GPU- and FPGA-based related works show that the intrusion detection problem for cybersecurity remains open. In addition, the combination of the complex and extensive nature of the supply chain with cybersecurity leads to a highly-demanding problem that needs to be solved in the modern industry environment. Hence, this is the first work that (i) employs a network security application (such as a network intrusion detection system) as a fundamental part of a supply chain infrastructure, (ii) reduces the infrastructure costs by using low-cost, power-efficient hardware accelerators, such as reconfigurable hardware, and (iii) sets free supply chain infrastructure from cybersecurity checks, offering the supply chain assets higher levels of processing freedom.

## 3. Intrusion Detection and Pattern Matching

One of the key functionalities of our engine is the exact string and binary signature matching, which is the approach of locating the occurrence of a simple string/binary pattern into another one. The exact string matching problem is to find all sub-strings in the input text that are exactly the same as the pattern. For example, the pattern exact matches the string exactly inside an input text. Each data stream (packet flow) is matched against large sets of strings (rules) with a single pass over the input bytes. This is achieved by utilizing the Aho–Corasick string matching algorithm. The main principle of the Aho–Corasick algorithm is that all patterns, fixed strings, and binary signatures, are compiled into a single DFA. Each byte of the input data stream moves the current DFA state to the next correct state. A match is achieved when a state, marked as final during the construction of the DFA, is encountered as the current state. The process of advancing through the state machine using one byte of the input stream at a time continues until the whole input is consumed. The Aho–Corasick algorithm seems to be a perfect fit for our application, since no backtracking on the input data is required and the process of acquiring the next DFA state lacks control flow instructions.

### 3.1. Multi-Pattern Matching with Aho–Corasick

Aho–Corasick is one of the most widely used algorithms that performs simple string pattern matching [47] and is considered as the best option for multiple pattern searching, since it matches all signatures simultaneously in a single pass. This simultaneous matching can be achieved when the set of patterns is being preprocessed. In the preprocessing phase, an automaton is built, which will be used eventually in the matching phase. Additionally, each character of the text will be processed only one time during the matching phase. The Aho–Corasick algorithm has the property that, theoretically, the processing time does not explicitly depend on the number of patterns. Letting $P = p_1 p_2 \ldots p_n$ be the patterns to be searched inside a text $T = t_1 t_2 \ldots t_m$ (with lengths $n$ and $m$, accordingly), both sequences of characters form a finite character set $\Sigma$. The complexity of the algorithm is linear in the pattern length $\nu$, plus the length of the given text $\mu$, plus the number of output matches.

Given a set of patterns, the algorithm constructs a pattern matching machine that matches all patterns in the text at once, one byte at a time. Each processing action of the automaton accepts an input event. The very first action starts with the initial state, represented by zero. Each action that accepts an input event moves the current state to the next state, based exclusively on that input. There are three distinct functions: (a) a `goto` function, (b) a `failure` function, and (c) an `output` function. According to the input event, one function is being triggered. Figure 2 presents an example of these functions for the set of patterns, i.e., {he, she, his, hers}.

The `goto` function (Figure 2a) determines if a state transition can be performed, based on the current state and the ASCII value of the input character. If the input character matches one of the transitions starting from the current state, then the state pointed to by this transition becomes the next state. Otherwise, the next state is resolved by the `failure` function $f(i = current\ state)$. For example (based on Figure 2a), the edge labeled $h$ from 0 to 1 indicates that $goto(0, h) = 1$, while the absence of an arrow for $a$ indicates failure. The `failure` function either drives a transition to one or more intermediate states, or to

the initial state (the one that is represented with 0 in the `goto` graph). After each state transition, the algorithm checks the `output` function $output(i = current\ state)$ in order to determine if the pattern matches a sub-string of the text $T$. This procedure continues and terminates at the end of the input text $T$. Since failed transitions may not consume any input—the so-called $\epsilon$-transitions—the produced automaton is non-deterministic (NFA). Additionally, the `failure` function can result in numerous state transitions for a single input character. In this way, the matching operation might require the exploration of multiple paths before the actual match of a pattern. A revised version of the traditional Aho–Corasick exists and replaces all failure transitions in order to avoid the performance loss, when the patterns' sizes are large. The new automaton that is produced is called "deterministic finite automaton" (DFA) and provides one transition per state and input character. Despite this approach requiring more memory than the previous one, it is more efficient in terms of processing throughput. The achieved complexity of this approach is $O(n)$.

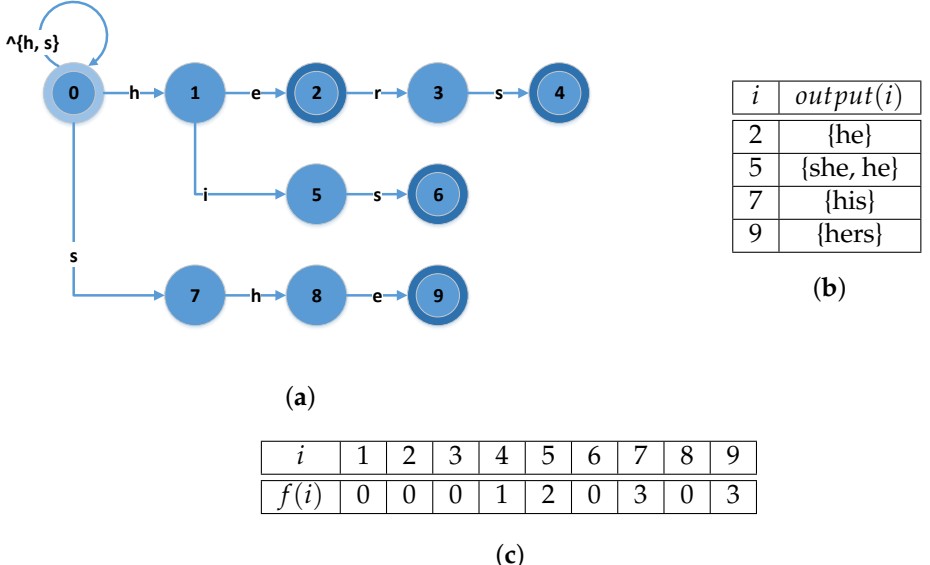

(a)

| $i$ | $output(i)$ |
|---|---|
| 2 | {he} |
| 5 | {she, he} |
| 7 | {his} |
| 9 | {hers} |

(b)

| $i$ | 1 | 2 | 3 | 4 | 5 | 6 | 7 | 8 | 9 |
|---|---|---|---|---|---|---|---|---|---|
| $f(i)$ | 0 | 0 | 0 | 1 | 2 | 0 | 3 | 0 | 3 |

(c)

**Figure 2.** An example of multi-pattern matching with Aho–Corasick. (**a**) The Aho–Corasick `goto` function, presented as a state machine for the patterns she, he, hers, his. (**b**) The `output` function, indicating the final state of each pattern in the given set. (**c**) The Aho–Corasick `failure` function for the given pattern set.

### 3.2. Intrusion Detection with Snort

Snort [48] is one of the most popular intrusion detection and prevention systems (IDS/IPSs). It is open-source and enables real-time packet logging and traffic analysis, creating alerts when malicious network activities are matched against the signatures used. Aiming to identify an intrusion attempt, Snort monitors the incoming network traffic packets by inspecting packet headers and application layer data. If the packet content matches a specific pattern that is known to signify malicious activity, then an alert is raised. As network traffic speeds and volumes are continually increasing, the need for real-time, high-speed network inspection systems emerges. Thus, Snort, from its very beginning, started investing in the development of a pattern matching engine able to cope with gigabit network performance and significantly long rule sets.

As described above, Aho–Corasick is a very efficient algorithm that offers simultaneous multi-pattern matching and is a pivotal pattern matching algorithm that Snort implements, mainly due to its performance capabilities. The Aho–Corasick state machine is a finite state machine, which is a representation of every possible state of a system, packed with a collection of acceptable state transitions for the system. Snort implements the Aho–Corasick state machine with a deterministic finite automaton (DFA), since DFAs

enable the correct new state within a single transition, in contrast to non-deterministic finite automata (NFA) that may require multiple transitions before ending up in a new state [49]. The way that state transitions are represented in the state table can affect the memory footprint and the processing performance. Similarly, the caching properties of an algorithm can significantly affect its performance (i.e., fitting a state table into a single cache significantly accelerates the processing). Thus, Snort processes patterns into pattern groups and uses an optimized version of the state table storage format to reduce excess cache misses when possible.

### 3.3. Aho–Corasick Optimization

Aiming to take advantage of Snort's performance characteristics and the performance capabilities of a small, inexpensive, and independent FPGA, we redesign the Aho–Corasick version used by Snort, while we port Snort's source code to enable network intrusion detection on a PYNQ Z1 device.

In most implementations, the state machine (DFA) is constructed as a tree with each node containing information about the state it represents, as well as various metadata. Since complex data structures using pointers are not an appropriate fit for achieving high performance on our platform, we choose to represent the DFA as a serialization of the state machine tree to a single-dimensional integer array. In order to make the process of constructing this array easy to follow, we will describe the procedure using a two-dimensional array as the DFA representation. We will use the tree of Figure 2, produced by the patterns {he, she, his, hers}, as an example.

During the bootstrap phase, the various signatures are processed by a simple parser. The purpose of the parser is to identify the signatures and process the binary notations, if any. When all the available patterns are gathered and processed, they are compiled into a single Aho–Corasick DFA, constructed as a tree. The next step is to serialize the produced tree as a two-dimensional integer array. This array will have 256 columns, which represent the size of the ASCII set, and N rows, where N is the number of the states in the DFA. Each row represents a DFA state and each cell contains the number of the next valid transition, corresponding to the ASCII character that the cell represents. An example of this array can be found in Figure 3.

**ASCII set**

| States | 0 | ... | e 101 | ... | h 104 | i 105 | ... | r 114 | s 115 | ... | 255 |
|---|---|---|---|---|---|---|---|---|---|---|---|
| 0 | 0 | 0 | 0 | 0 | 1 | 0 | 0 | 0 | 7 | 0 | 0 |
| 1 | 0 | 0 | -2 | 0 | 1 | 5 | 0 | 0 | 7 | 0 | 0 |
| 2 | 0 | 0 | 0 | 0 | 1 | 0 | 0 | 3 | 7 | 0 | 0 |
| 3 | 0 | 0 | 0 | 0 | 1 | 0 | 0 | 0 | -4 | 0 | 0 |
| 4 | 0 | 0 | 0 | 0 | 8 | 0 | 0 | 0 | 7 | 0 | 0 |
| 5 | 0 | 0 | 0 | 0 | 1 | 0 | 0 | 0 | -6 | 0 | 0 |
| 6 | 0 | 0 | 0 | 0 | 8 | 0 | 0 | 0 | 7 | 0 | 0 |
| 7 | 0 | 0 | 0 | 0 | 8 | 0 | 0 | 0 | 7 | 0 | 0 |
| 8 | 0 | 0 | -9 | 0 | 1 | 5 | 0 | 0 | 7 | 0 | 0 |
| 9 | 0 | 0 | 0 | 0 | 1 | 0 | 0 | 3 | 7 | 0 | 0 |

**Figure 3.** The state table produced by the serialization of the Aho–Corasick DFA as a two-dimensional integer array. Negative values indicate final states.

To traverse the serialized DFA tree, the string matching task starts from state 0 (row 0) and selects the appropriate column, according to the ASCII value of the first character of the input. In this cell, it finds the next valid state, which is located in another row of the array. Then, it fetches the next character from the input and moves to the cell pointed to by the row given in the previous step and the column given by the ASCII value of the current

character. The final states in the array are annotated with a negative sign. When the task hits a negative state, we know that a match has been successfully found. Then, the search is continued using its absolute value for the next step. The fail states either point the matcher to a previous valid state or to the initial state 0.

In practice, this array is single-dimensional and all the rows that we mentioned earlier are concatenated. Since the size of every row is 256 integers, the `goto` function traverses the array as follows: `state = dfa[state * 256 + char_ASCII_value]`.

## 4. System Architecture

This section presents the implementation of the supply chain cybersecurity framework on reconfigurable technology. First, the reconfigurable architecture of the Aho–Corasick algorithm and its integration with the official Snort IDS is presented. Next, the implemented system is mapped into a supply chain and its intrusion detection service is evaluated.

### 4.1. Reconfigurable Aho–Corasick Architecture

As presented in Section 2, the Aho–Corasick algorithm has been mapped on various HPC reconfigurable platforms. This work offers the first implementation of the Aho–Corasick algorithm on a modern, inexpensive, and low-resources FPGA platform, i.e., PYNQ Z1. The goal of mapping Aho–Corasick on such platform type is the easy integration of the system into a complex environment such as supply chains. Additionally, this implementation focuses on low-cost intrusion detection systems for such huge environments. Last, the goal of mapping the Aho–Corasick algorithm on a reconfigurable platform is based on the fact that FPGAs can offer a high-performance solution for the pattern matching problem.

The mapped Aho–Corasick architecture consists of a pipeline architecture, as presented in Figure 4. The system input (i.e., the incoming network packets) is streamed and stored by a Cortex A9 processor to the onboard DDR memory. Additionally, the onboard DDR memory stores the predefined structure, i.e., a tree-based data structure, of the rules that defines the "suspicious" network packets. Then, the processor passes to the reconfigurable device the parameters that are used for running the Aho–Corasick algorithm (e.g., the number of stored rules and the the input data size that is going to be processed).

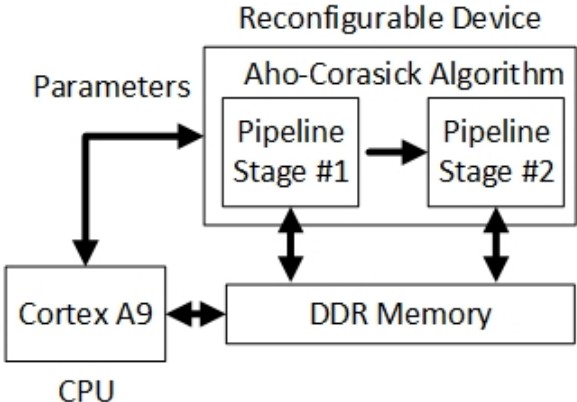

**Figure 4.** Reconfigurable Aho–Corasick architecture.

The implemented architecture consists of two high-level pipeline stages, which are internally split into two lower-level pipeline stages. The processing results return from the second pipeline stage and they are stored back to the DDR memory, where the Cortex processor has access and retrieves them. The pipeline stages implement a low-level finite state machine (FSM), which is responsible to move into stored tree structure according to the streaming input data. The movement on the prestored tree structure produces a final score, which actually defines if the processed data includes problematic information data, and the IDS solution is activated.

The architecture of the Aho–Corasick algorithm is implemented with Vivado HLS and it is mapped on the reconfigurable platform as a Vivado IP module. Its operating frequency is 410 MHz with really low resource utilization, i.e., it uses up to 10% of the available LUT resources. The IP is mapped on the reconfigurable platform of a Zynq-7000 SoC.

### 4.2. Snort-Based Reconfigurable Solution

This section describes the integration of the reconfigurable Aho–Corasick module with the official Snort implementation, which were finally mapped on a PYNQ Z1 board. PYNQ Z1 is a platform built on top of Xilinx Zynq SoC technology that allows developers to exploit the programmable logic of the board directly from Python applications. The advantages of this device is its high-performance capabilities, since it is significantly low-cost and offers the opportunity to designers to build real-time systems.

Figure 5 presents the architecture of the reconfigurable IDS node. In more detail, Snort runs on the integrated Cortex A4 CPU. The algorithm collects the incoming data packets (i.e., Ethernet network packets) and stores them into the PYNQ Z1 DDR memory. The reconfigurable device reads the incoming packets and processes them based on the parameters from the CPU. The results from the Aho–Corasick module are stored back to the DDR and they are retrieved by the software-based Snort. If there is an intrusion alert, a message with the corresponding information is passed through Ethernet to the supply chain server. The presented system architecture was fully implemented with Xilinx Vitis 2020.1 and was mapped on a PYNQ Z1 device.

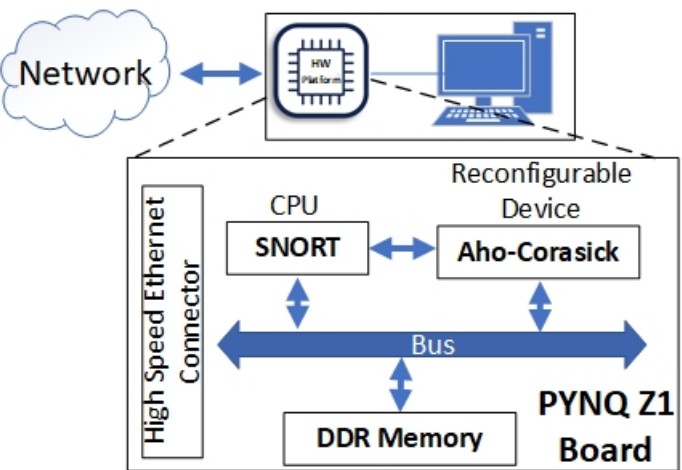

**Figure 5.** Reconfigurable Aho–Corasick architecture is integrated with the official Snort IDS on a PYNQ Z1 board.

### 4.3. Supply Chain Integration

The PYNQ Z1 board offers different interconnection capabilities; (i) USB and (ii) two gigabit Ethernet ports (1 Gbps each). We choose to integrate the board that maps the reconfigurable Aho–Corasick design into a supply chain framework using the two available Ethernet ports. Specifically, our proposed framework places a PYNQ Z1 board in front of each supply chain asset. One of the two Ethernet board ports is connected to the network receiving the incoming packets, which are directed to the protected asset. Next, the incoming packets are processed on the PYNQ platform and they are forwarded through the second Ethernet port, which connects directly the PYNQ board with the supply chain asset. The proposed framework of integrating our components into a supply chain structure is presented in Figure 6. Based on the proposed framework, the PYNQ components need to be placed as intermediate nodes between the assets' incoming network connection and the supply chain assets. This solution covers the supply chain structure from both the external (i.e., external network connection) and the internal (i.e., other supply chain components) cybersecurity threats.

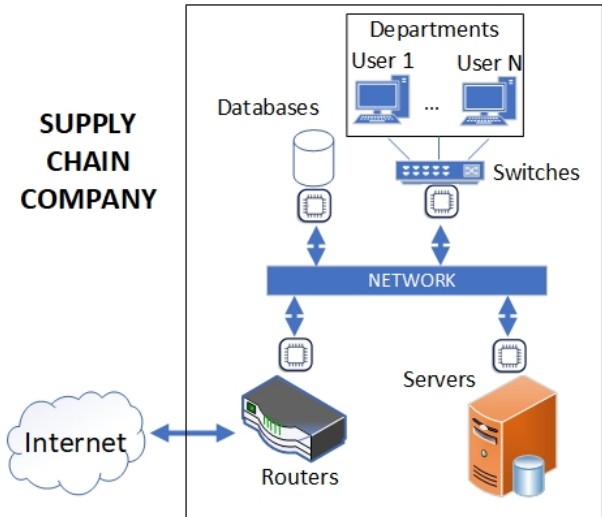

**Figure 6.** PYNQ Z1 integration into supply chain framework.

## 5. System Performance

The performance of the proposed framework was evaluated against several network traffic datasets over an emulated supply chain infrastructure. Specifically, the PYNQ board hosting the reconfigurable Aho–Corasick implementation was connected in between a local PC (referred to as the asset) using one Ethernet port and an emulated "external" network using the second port. Other network PCs communicated with the protected asset directly through the network. Based on the test scenario, the local PCs sent either "invalid/suspicious" packets, which triggered alerts by the PYNQ board, or "valid/benign" network packets, which arrived normally to the protected asset. The PYNQ board, which hosted the online IDS module, checked all the incoming packets and it either sent log alert messages to the server console when an issue appeared or allowed the packets to continue onto the asset. Four different real-world network traffic datasets (i.e., [50–53]) were used to evaluate the proposed framework. The datasets consist of various network packet types (i.e., TCP, UDP, and ICMP) with fluctuating sizes (64 bytes up to 1500 bytes). All datasets are related to Industrial IoT (IIoT) for evaluating the fidelity and efficiency of cybersecurity applications. The structure of the emulated supply chain, which was used for system evaluation, is presented in Figure 7.

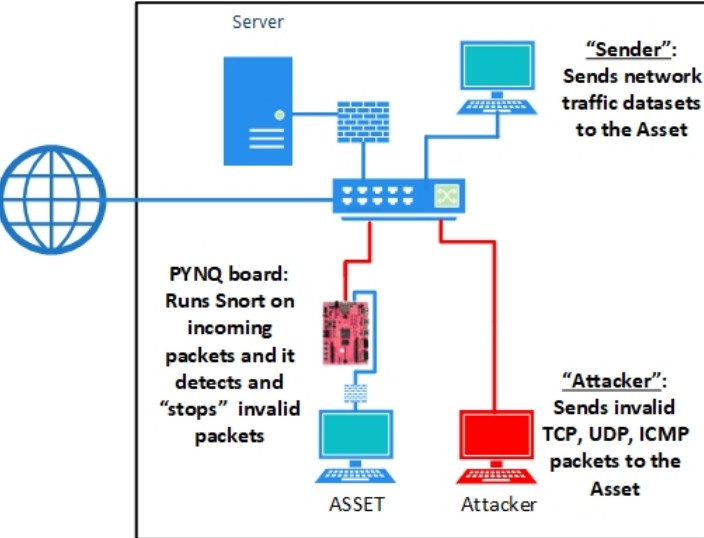

**Figure 7.** "Virtual" supply chain test infrastructure.

*5.1. Performance*

The performance of the reconfigurable IDS module was measured based on two different scenarios. The first scenario focused on the performance comparison of the reconfigurable IDS system vs. the best optimized software-based IDS solution mapped on the same platform (i.e., PYNQ board), without taking into account the network delay. The second scenario focused on the PYNQ board performance when it was integrated into an emulated supply chain infrastructure and processed data concurrently from different assets (i.e., using a combination of "invalid/suspicious" and "valid/benign" network packets).

The first scenario moved on the performance evaluation of the reconfigurable PYNQ IDS system without taking into account any network delay. Specifically, Snort was mapped on a single PYNQ board consuming the input network traffic from local pcap files. The data were preprocessed by the official software solution on the PYNQ's Cortex A9 processor and they were stored temporarily into the onboard DRAM. Next, they were streamed to the reconfigurable device, where they were processed, while the corresponding results, i.e., intrusion detection alerts or clearance, were streamed through local DRAM back to software. Lastly, depending on the results from the reconfigurable device, the software system either sent the corresponding "alert" messages to the supply chain admin or forwarded the network packets to the asset. The performance of the reconfigurable module was compared against the performance of the best optimized Snort solution under exactly the same conditions, i.e., the Cortex A9 processor was used for running the most optimized Snort software solution.

Table 2 shows the performance results achieved for the different input datasets. The presented performance results are the average values obtained after several runs. Based on these results, the reconfigurable solution of the Snort system, i.e., Snort running on the Cortex processor while Aho–Corasick is mapped on the FPGA (reconfigurable) device, outperforms the best optimized software solution by a factor of up to $1.4\times$. It is important to mention that both software implementations, i.e., the official Snort system and the preprocessing steps for the reconfigurable solution, were fully optimized. On the other hand, as the resource utilization for the reconfigurable architecture is really low and the mapped architecture used only up to 10% of the available PYNQ LUTs, the proposed system can be easily parallelized and can process parallel network streaming connections concurrently, increasing the performance achievements of the reconfigurable system.

**Table 2.** Performance for the official fully-optimized Snort implementation framework vs. the reconfigurable Snort implementation on a PYNQ platform.

| Datasets | Official Software Snort (s) | Reconfigurable Snort Implementation (s) | Acceleration |
|---|---|---|---|
| [50] | 2.77 | 2.38 | $1.2\times$ |
| [52] | 33.29 | 26.71 | $1.2\times$ |
| [51] | 108.94 | 79.39 | $1.4\times$ |
| [53] | 152.52 | 118.13 | $1.3\times$ |

The second scenario focused on the network packet throughput rates that can be achieved by the proposed reconfigurable system in real-world conditions (e.g., fluctuating packet sizes). During this scenario, the PYNQ board received data from various supply chain assets and concurrently processed attacks with "invalid/suspicious" network packets. Specifically, the "sender", as shown in Figure 7, started transmitting over the supply chain network one of the four previous scenario datasets. Concurrently, an "attacker" started sending "invalid/suspicious" packets, which were checked and stopped by the reconfigurable Snort system. The performance of the implemented system was evaluated based on the network packet throughput rate (average values), which is presented in Table 3. According to the results achieved, the packet processing rate of the optimized software solution for small-sized packet datasets (i.e., [50,52]) is close to the reconfigurable system,

without offering any significant improvement. These results were expected, since those datasets contain a large number of small-sized packets that lead to a large number of interrupts that affect the overall processing rate. On the contrary, the reconfigurable system offers higher processing rates for datasets with mainly large-sized packets (i.e., [51,53]), as shown in the table below. It is crucial to note that the particular MPSoC used supports greater parallelization in the way with which data can be supplied to the FPGA for IDS processing. Hence, the measured figure of $1.4\times$ can in fact be doubled by simply instantiating twice the existing design within the FPGA fabric, leading to an acceleration of $2.8\times$. Furthermore, a simple modification to the current design could dedicate all I/O ports to data supply for processing, thereby quadrupling the achieved performance and leading to an acceleration of $5.6\times$. This is an estimate that can be attained in a straightforward way and with minimal effort due to the four parallel DMA connections between the PYNQ processor and reconfigurable device.

**Table 3.** Network packet processing rate comparison between the official fully-optimized Snort implementation framework vs. the reconfigurable Snort implementation on a PYNQ platform.

| Datasets | Official Software Snort (Packets/s) | Reconfigurable Snort Implementation (Packets/s) | Packet Processing Rate Optimization |
|:---:|:---:|:---:|:---:|
| [50] | 6830 | 6147 | $0.9\times$ |
| [52] | 13,770 | 13,365 | $1.0\times$ |
| [51] | 17,213 | 22,098 | $1.3\times$ |
| [53] | 5555 | 7980 | $1.4\times$ |

*5.2. Energy Consumption*

This section analyzes the energy consumption advantages of the proposed reconfigurable-based IDS solution vs. the official software implementation on a high-end server. Specifically, very recently, Shehabi et al. [54] presented that a modern high-end single-socket server consumes 118 Watts per hour and a two-socket server consumes 365 Watts per hour. With respect to the reconfigurable hardware solution proposed here, Table 4 presents the power consumption of the PYNQ Z1 board while breaking it down into its individual contributors. It can, therefore, be calculated that the complete implementation setup, comprising all listed PYNQ Z1 board components, leads to an overall energy consumption of 13.82 Watts per hour. Taking into account that the performance acceleration of the PYNQ-based solution is between $1\times$ and $1.4\times$, then the corresponding energy consumption of the proposed system is up to almost $12\times$ lower than the single-socket server, which constitutes a notable reduction.

**Table 4.** PYNQZ1 power supplies [55].

| Supply | Circuits | Current (Max) |
|:---:|:---:|:---:|
| 3.3 V | FPGA I/O, USB ports, Clocks, Ethernet | 1.6 A |
| 1.0 V | FPGA, Ethernet Core | 2.6 A |
| 1.5 V | DDR3 | 1.8 A |
| 1.8 V | FPGA Ethernet I/O, USB Controller | 1.8 A |

## 6. Discussion

Supply chains are systems that link businesses to their suppliers, allowing them to produce and distribute goods and products. Supply chain management is based on the flow of data, information, resources, and materials in order to deliver the best product and service to all stakeholders. Cybersecurity and risk management have always been vital for the efficient flow of any business; nevertheless, there is an increasing amount of

strain exercised on both aspects regarding SC cybersecurity. Hence, organizations within the supply chain need to adopt active and focused measures, i.e., mitigation strategies, in order to avoid the cybersecurity consequences, i.e., crime-related delays, data breaches, and financial losses [56]. One of the most efficient solutions to this problem is the placement of technologies with strong security measures into supply chain infrastructures. Our results show that cheap and small multi-processor system-on-chip (MPSoC) platforms can be easily integrated with supply chain infrastructures and have the cybersecurity workload offloaded onto them. Our system accelerates a widely-used IDS, namely, Snort, by mapping it on a small and cheap reconfigurable platform, i.e., PYNQ Z1. The system is integrated with an SC infrastructure and its performance is evaluated with real-life workloads. The performance results with input datasets from complex supply chain networks show that the implemented reconfigurable system can offer real-time processing.

The technological contributions of this work can be grouped mainly on two fronts. First, they illustrate the advantages that FPGA-based solutions posses over the conventional CPU-based solutions. One such great advantage is that heavy cybersecurity workloads can be *offloaded completely* from the main supply chain components onto dedicated peripherals such as the FPGAs. Moreover, using lightweight and dedicated hardware for security purposes reduces the overall size of the framework resources and, subsequently, their cost. The size of the proposed FPGA boards, i.e., PYNQ Z1, is really small and they can be easily integrated with, and placed into, infrastructures, such as a server room, that pose strict specifications on the allowable size of resources used for IDS and security purposes. In addition, the cost of the presented solution is really low, i.e., each board costs about USD 300, compared to the much higher cost of modern servers, which are used for cybersecurity workloads. Furthermore, the proposed solution takes advantage of one of the main characteristics of reconfigurable-based platforms, i.e., the low-energy consumption. Such hardware offers much higher energy efficiency over any other proposed solution, i.e., GPUs or high-end servers, used in the context of IDS in SC systems, such as shown in the previous section. Finally, the proposed reconfigurable hardware-based IDS architecture allows dynamic and continuous updating. Specifically, the FPGA-based system can easily add new rules dynamically by simply uploading, storing, and updating the corresponding dedicated internal FPGA memories without interrupting the system operation.

In addition, this work reveals the potential that the FPGA-based solutions posses over the GPU-based IDS solutions. The FPGA devices used for mapping the proposed architecture have a lower cost, i.e., about USD 300, compared to that for GPUs that can offer the same performance figures. In addition, the FPGA offers higher flexibility over GPU platforms as far as their internal architecture is concerned, which leads to more efficient system mapping and to better performance. Additionally, the FPGA devices used, i.e., PYNQ boards, seem to have higher market availability, in order to cover the needs of huge supply chain infrastructures, than the corresponding low-cost GPU devices.

Finally, as subsequent research activities, we decided to focus on three main directions, i.e., (i) to introduce a greater number of parallelization levels on the reconfigurable device, i.e., parallel use of memory ports or increasing the processed packets size, which will increase the processing rate of the incoming ethernet packets; (ii) the architecture can be further parallelized using independent PYNQ platforms, using more than just one PYNQ board, to map different rules concurrently, thereby offering higher levels of throughput and security; finally, (iii) integrate and evaluate a complete prototype comprising a collection of PYNQ boards on a real-life supply chain infrastructure offered within the context of the EU-research project CYRENE [1].

## 7. Conclusions

This work presents the very first implementation of a hybrid security scheme comprising two parts, i.e., the CPU-based Snort algorithm combined with a novel reconfigurable hardware-based Aho–Corasick intrusion detection system. The proposed system is mapped on an inexpensive and low-resources reconfigurable device, i.e., PYNQ Z1, which has been

integrated with and tested using a "virtual" supply chain infrastructure similar to the end system that is going to be provided in the context of EU-funded project CYRENE [1].

The proposed work advances the cybersecurity capabilities of supply chains in a range of different ways. First, the presented system offers real-time cyber threat monitoring and intrusion detection over complex supply chain infrastructures. Moreover, the integration of the described modules into supply chain infrastructures releases other supply chain components from cybersecurity tasks, leading to optimized supply chain infrastructures. Additionally, the proposed low-cost reconfigurable system offers performance acceleration for one of the most typical intrusion detection systems, i.e., Snort. Crucially, the improved performance is accompanied by a significant reduction in energy consumption, i.e., we improve on the performance at a significant gain in power and energy consumption. Finally, the results from the presented work demonstrate that, indeed, small and reconfigurable devices can be easily integrated into complex infrastructures, such as supply chains, enhancing the level and quality of the cybersecurity services that they can deliver.

**Author Contributions:** Methodology, D.D., G.C. and K.G.; Project administration, K.G. and S.I.; Software, D.D.; Hardware, G.C. and K.G.; Supervision, S.I.; Writing—original draft, E.P., D.D. and G.C.; Writing—review & editing, E.P., G.C. and K.G. All authors have read and agreed to the published version of the manuscript.

**Funding:** This research was funded by the European Horizon 2020 Research and Innovation Programme CYRENE under Grant Agreement No 952690. Also, this work was funded by the European Horizon 2020 Innovation Programme EnerMan under grant agreement No 958478.

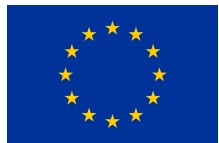

**Institutional Review Board Statement:** Not applicable.

**Informed Consent Statement:** Not applicable.

**Data Availability Statement:** Publicly available datasets were analyzed in this study. These datasets can be found in the next web pages: https://www.malware-traffic-analysis.net/ (accessed on 1 March 2022), https://www.ll.mit.edu/r-d/datasets/1999-darpa-intrusion-detection-evaluation-dataset (accessed on 3 March 2022), https://research.unsw.edu.au/projects/toniot-datasets (accessed on 8 April 2022) and https://github.com/nccgroup/Cyber-Defence/tree/master/Intelligence/Honeypot-Data/2020-F5-and-Citrix (accessed on 13 April 2022).

**Acknowledgments:** The authors would like to thank the anonymous reviewers and the editor. This work was supported by the projects CYRENE and EnerMan funded by the European Commission under Grant Agreements No 952690 and No 958478.

**Conflicts of Interest:** The authors declare no conflict of interest.

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
