# Peer review of "The Diversification and Enhancement of an IDS Scheme for the Cybersecurity Needs of Modern Supply Chains"

_electronics, doi:10.3390/electronics11131944_

Round 1

Reviewer 1 Report

The article "The Diversification and Enhancement of an IDS Scheme for the Cybersecurity Needs of Modern Supply Chains" is an interesting study. But there are some concerns:

1. Please, refer to recent articles on 2021 and 2022 in the introduction to state the issue.

2. In the literature, refer to the conducted studies in the last two years (2021-2022).

3. In the literature section, a comparison table of previous studies should be presented and the studies should be compared.

4. In the methodology section, more relations should be provided regarding the method.

5. Result and discussion section should be added in the article to compare the research results with previous studies. Also, the study results should be analyzed.

6. Please, provide further analysis of tables and figures.

7. Please, improve the quality of figures.

8. The article should be examined for grammatical problems.

9. At the end of the Conclusions section, first mention the limitations of the study and make research suggestions based on the limitations.

Reviewer 2 Report

Despite the importance of supply chains in terms of socioeconomics, security officers and operators lack an easy and integrated way to defend their interconnected vital infrastructures from cyber-attacks. Malicious activity detection solutions and approaches are constantly being explored and offered in the supply chain environment. This paper describes the development of a low-cost reconfigurable intrusion detection system that may be quickly integrated into supply chain networks while maintaining high security standards. By unloading supply chain components from security check workloads, the proposed system provides real-time cyber security intrusion detection through high-speed networks and services. This method employs a unique framework that implements the Aho-Corasick algorithm on a Multi-Processor System-on-Chip platform's reconfigurable fabric, allowing for concurrent matching of numerous network packet patterns. The Multi-Processor System-on-Chip proof-of-concept system surpasses the present software-based official solution while relieving supply chain nodes of the burden of high-workload cyber security checks, according to the initial performance evaluation.

In this form, the version is good, but the majority of the latest references are absent. The article's English is weak, and several sentences are very long. There are also no comparisons between approaches. Please update the acknowledgement and funding statements, as well. In addition, edit several references and add some key comments concerning the work's benefits and drawbacks. This manuscript can be accepted for publication after a critical revision. My other suggestion is major and important change, kindly cited the hottest references. 

Reviewer 3 Report

In general I find the paper interesting and worthy publishing. However, I feel that the paper is written for a specialist who is an expert in supply chain cybersecurity and I worry that there are not too many such experts among Electronics readers. I think that the authors should put more effort to explain what, how and why they do what they do. For instance:

Why they consider GPU or FPGA implementation of the system. IDS/IPS can run almost on anything. Snort for instance could run simply on a standard PC. Why GPU or FPGA is necessary?

The performance improvement shown in Tables 1 and 2 seems rather moderate. Could the authors explain why they consider such gain a significant improvement of performance? I guess (authors note it too) that parallelising the processes would most likely provide better gains.

What is the difference between a 'supply chain network' studied here and a standard IP network? (LAN + WAN).

Also:

Line 63: what is ''really impressive performance''? The authors should use either a metric here or compare with some reference.

Round 2

Reviewer 1 Report

The authors modified the article well.

Reviewer 2 Report

Now, paper can be accepted for publications.